# Evaluation of mitochondrial DNA copy number estimation techniques

Ryan J. Longchamps[1], Christina A. Castellani[1], Stephanie Y. Yang[1], Charles E. Newcomb[1], Jason A. Sumpter[1], John Lane[2], Megan L. Grove[3], Eliseo Guallar[4], Nathan Pankratz[2], Kent D. Taylor[5], Jerome I. Rotter[5], Eric Boerwinkle[3,6], Dan E. Arking[1]*

**1** Department of Genetic Medicine, McKusick-Nathans Institute, Johns Hopkins University School of Medicine, Baltimore, MD, United States of America, **2** Department of Laboratory Medicine and Pathology, University of Minnesota Medical School, Minneapolis, MN, United States of America, **3** Human Genetics Center, Department of Epidemiology, Human Genetics, and Environmental Sciences, School of Public Health, The University of Texas Health Science Center at Houston, Houston, TX, United States of America, **4** Department of Epidemiology and the Welch Center for Prevention, Epidemiology and Clinical Research, Johns Hopkins Bloomberg School of Public Health, Baltimore, MD, United States of America, **5** LABioMed and Department of Pediatrics, at Harbor-UCLA Medical Center, Institute for Translational Genomics and Population Sciences, Torrance, CA, United States of America, **6** Human Genome Sequencing Center, Baylor College of Medicine, Houston, TX, United States of America

* arking@jhmi.edu

**Data Availability Statement:** Data from this study are available upon request as these data contain potentially identifying and sensitive patient information. Individuals who wish to access these

## Abstract

Mitochondrial DNA copy number (mtDNA-CN), a measure of the number of mitochondrial genomes per cell, is a minimally invasive proxy measure for mitochondrial function and has been associated with several aging-related diseases. Although quantitative real-time PCR (qPCR) is the current gold standard method for measuring mtDNA-CN, mtDNA-CN can also be measured from genotyping microarray probe intensities and DNA sequencing read counts. To conduct a comprehensive examination on the performance of these methods, we use known mtDNA-CN correlates (age, sex, white blood cell count, Duffy locus genotype, incident cardiovascular disease) to evaluate mtDNA-CN calculated from qPCR, two microarray platforms, as well as whole genome (WGS) and whole exome sequence (WES) data across 1,085 participants from the Atherosclerosis Risk in Communities (ARIC) study and 3,489 participants from the Multi-Ethnic Study of Atherosclerosis (MESA). We observe mtDNA-CN derived from WGS data is significantly more associated with known correlates compared to all other methods ($p < 0.001$). Additionally, mtDNA-CN measured from WGS is on average more significantly associated with traits by 5.6 orders of magnitude and has effect size estimates 5.8 times more extreme than the current gold standard of qPCR. We further investigated the role of DNA extraction method on mtDNA-CN estimate reproducibility and found mtDNA-CN estimated from cell lysate is significantly less variable than traditional phenol-chloroform-isoamyl alcohol ($p = 5.44 \times 10^{-4}$) and silica-based column selection ($p = 2.82 \times 10^{-7}$). In conclusion, we recommend the field moves towards more accurate methods for mtDNA-CN, as well as re-analyze trait associations as more WGS data becomes available from larger initiatives such as TOPMed.

data are welcome to contact the ARIC coordinating center at UNC (aricpub@unc.edu) or the MESA coordinating center (chsccweb@u.washington.edu).

**Funding:** This research was supported by grant R01HL131573 from the US National Institutes of Health (Longchamps, Castellani, Guallar, and Arking) and by grant P30AG021334 from the Johns Hopkins University Claude D. Pepper Older Americans Independence Center National Institute on Aging (Dr Arking). The Atherosclerosis Risk in Communities study has been funded in whole or in part with Federal funds from the National Heart, Lung, and Blood Institute, National Institutes of Health, Department of Health and Human Services, under Contract nos. (HHSN268201700001I, HHSN268201700002I, HHSN268201700003I, HHSN268201700005I, HHSN268201700004I). The Multi-Ethnic Study of Atherosclerosis is supported by contracts HHSN268201500003I, N01-HC 95159, N01-HC-95160, N01-HC-95161, N01-HC-95162, N01-HC-95163, N01-HC-95164, N01-HC-95165, N01-HC-95166, N01-HC-95167, N01-HC-95168 and N01-HC-95169 from the National Heart, Lung, and Blood Institute, and by grants UL1-TR-000040, UL1-TR-001079, and UL1-TR-001420 from the National Center for Advancing Translational Sciences (NCATS). Dr. Rotter and Dr. Taylor's efforts were supported in part by the National Center for Advancing Translational Sciences, CTSI grant UL1TR001881, and the National Institute of Diabetes and Digestive and Kidney Disease Diabetes Research Center (DRC) grant DK063491 to the Southern California Diabetes Endocrinology Research Center. Funding for SHARe genotyping was provided by National Heart, Lung, and Blood Institute contract N02-HL-64278. The funding organizations had no role in the design and conduct of the study; collection, management, analysis, and interpretation of the data; preparation, review, or approval of the manuscript; and decision to submit the manuscript for publication.

**Competing interests:** The authors have declared that no competing interests exist.

## Introduction

Mitochondrial dysfunction has long been known to play an important role in the underlying etiology of several aging-related diseases, including cardiovascular disease (CVD), neurodegenerative disorders and cancer[1]. As an easily measurable and accessible proxy for mitochondrial function, mitochondrial DNA copy number (mtDNA-CN) is increasingly used to assess the role of mitochondria in disease. Several population-based studies have shown higher levels of mtDNA-CN to be associated with decreased incidence for CVD and its component parts: coronary artery disease (CAD) and stroke[2,3]; neurodegenerative disorders such as Parkinson's and Alzheimer's[4,5]; as well as several types of cancer including breast, kidney, liver and colorectal[6–8]. Furthermore, mtDNA-CN measured from peripheral blood has consistently been shown to be higher in women, decline with age, and correlate negatively with white blood cell (WBC) count[9–11].

Although the mtDNA-CN field is relatively young, the number of publications has been steadily increasing at an average rate of 12% per year since 2015[12]. However, there has yet to be a rigorous examination of the various methods for measuring this novel phenotype and the factors which may influence its accurate estimation. Without such an examination, studies may be severely underestimating or misrepresenting the relationship of mtDNA-CN with their traits of interest.

Quantitative real-time PCR (qPCR) has been the most widely used method for measuring mtDNA-CN, partly due to its low cost and quick turnaround time. However, recent work has demonstrated the feasibility of accurately measuring mtDNA-CN from preexisting microarray, whole exome sequencing (WES) and whole genome sequencing (WGS) data[2,10,13]. With these advances, it is important for the field to evaluate these methods in the context of the current gold standard.

In addition to the method for determining mtDNA-CN, it is important to consider the impact of DNA extraction method on mtDNA-CN, particularly due to the small size and circular nature of the mitochondrial genome. Previous research has shown organic solvent extraction is more accurate than silica-based methods at measuring mtDNA-CN, which is unsurprising as column kit parameters are typically optimized for DNA fragments ≥50 Kb [14]. However, as all DNA extraction methods have bias in the DNA which they target, measuring mtDNA-CN from direct cell lysate may prove to be a more accurate method.

In the present study, we assess the relative performance of various methods for measuring mtDNA-CN and the effects of DNA extraction on mtDNA-CN estimation accuracy. We leverage mtDNA-CN calculated across 4,574 individuals from two prospective cohorts, the Atherosclerosis Risk in Communities study (ARIC) and the Multi-Ethnic Study of Atherosclerosis (MESA). Using mtDNA-CN estimates calculated from qPCR, WES, WGS, and two microarray platforms–the Affymetrix Genome-Wide Human SNP Array 6.0 and the Illumina HumanExome BeadChip genotyping array–we compare associations for known correlates of mtDNA-CN including age, sex, white blood cell count, the Duffy locus and incident CVD to determine the optimal method for calculating copy number. We additionally determined the reproducibility of mtDNA-CN measurements *in vitro* from three separate DNA extraction methods: silica-based column selection, organic solvent extraction (phenol-chloroform-isoamyl alcohol), and measuring mtDNA-CN from direct cell lysis without performing a traditional DNA extraction. We hypothesized that mtDNA-CN calculated from WGS data would outperform other estimation methods and mtDNA-CN measured from direct cell lysate would be more accurate than traditional DNA extraction methods.

## Methods

### Study populations

The ARIC study recruited 15,792 individuals between 1987 and 1989 aged 45 to 65 years from 4 US communities. DNA for mtDNA-CN estimation was collected from different visits and was derived from buffy coat using the Gentra Puregene Blood Kit (Qiagen). Relevant covariates were derived from the same visit in which DNA was collected. Our analyses were limited to 1,085 individuals with mtDNA-CN data available across all four platforms performed within ARIC: Affymetrix Genome-Wide Human SNP Array 6.0, Illumina HumanExome BeadChip genotyping array, WES and WGS. Eighty-eight percent of our final ARIC participants were Black.

The MESA study recruited 6,814 individuals free of prevalent clinical CVD from 6 US communities across 4 ethnicities. Age range at baseline was 45 to 84 and the baseline exam occurred between 2000 and 2002. DNA for mtDNA-CN analyses was isolated from exam 1 peripheral leukocytes using the Gentra Puregene Blood Kit. Our analyses were restricted to 3,489 White and Black (36%) individuals with mtDNA-CN data available across the three platforms with mtDNA-CN data available at the time of analysis: qPCR, Affymetrix Genome-Wide Human SNP Array 6.0 and Illumina HumanExome BeadChip genotyping array. Exam 1 DNA for the exploratory dPCR pilot study was derived from packed red blood cells.

### Measurement of mtDNA-CN

**qPCR.** mtDNA-CN was determined using a multiplexed real time qPCR assay as previously described[11]. Briefly, the cycle threshold (Ct) value of a mitochondrial-specific (*ND1*) and nuclear-specific (*RPPH1*) target were determined in triplicate for each sample. The difference in Ct values (ΔCt) for each replicate represents a raw relative measure of mtDNA-CN. Replicates were removed if they had Ct values for *ND1* >28, Ct values for *RPPH1* >5 standard deviations from the mean, or ΔCt values >3 standard deviations from the mean of the plate. Outlier replicates were identified and excluded for samples with a ΔCt standard deviation >0.5. The sample was excluded if the ΔCt standard deviation remained >0.5 after replicate removal. We corrected for an observed linear increase in ΔCt value due to the pipetting order of each replicate via linear regression. The mean ΔCt across all replicates was further adjusted for plate effects as a random effect to represent a raw relative measure of mtDNA-CN.

**Microarray.** mtDNA-CN was determined using the Genvisis[15] software package for both the Affymetrix Genome-Wide Human SNP Array 6.0 and the Illumina HumanExome BeadChip genotyping array. A list of high-quality mitochondrial SNPs were hand-curated by employing BLAST to remove SNPs without a perfect match to the annotated mitochondrial location and SNPs with off-target matches longer than 20 bp. The probe intensities of the remaining mitochondrial SNPs (25 Affymetrix, 58 Illumina Exome Chip) were determined using quantile sketch normalization (apt-probeset-summarize) as implemented in the Affymetrix Power Tools software. The median of the normalized intensity, log R ratio (LRR) for all homozygous calls was GC corrected and used as initial estimates of mtDNA-CN for each sample.

Technical covariates such as DNA quality, DNA quantity, and hybridization efficiency were captured via surrogate variable analysis or principal component analysis as previously described[2]. Surrogate variables or principal components were applied to the BLAST filtered, GC corrected LRR of the remaining autosomal SNPs (43,316 Affymetrix, 47,512 Exome Chip).

These autosomal SNPs were selected based on the following quality filters: call rate >98%, HWE *p* value >0.00001, PLINK mishap for non-random missingness *p* value >0.0001,

association with sex $p$ value >0.00001, linkage disequilibrium pruning ($r^2$ <0.30), with maximal spacing between autosomal SNPs of 41.7 kb.

**WES.**   Whole exome capture was performed using Nimblegen's VChrome2.1 (Roche) and sequencing was performed on the Illumina HiSeq 2000. Sequence reads were aligned using Burrows-Wheeler Aligner (BWA)[16] to the hg19 reference genome. Variant calling, and quality control were performed as previously described[17]. mtDNA-CN was calculated using the mitoAnalyzer software package, which determines the observed ratios of sequence coverages between autosomal and mtDNA[18,19].

Due to large batch effects observed in our raw mtDNA-CN calls, alignment summary, insert size, quality score, base distribution, sequencing artifact and quality yield metrics were collected using Picard tools (version 1.87) to take into account differences in capture efficiency as well as sequencing and alignment quality[20]. Picard sequencing summary metrics to incorporate into our final model were selected through a stepwise backwards elimination model (S1 Table).

**WGS.**   Whole genome sequencing data was generated at the Baylor College of Medicine Human Genome Sequencing Center using Nano or PCR-free DNA libraries on the Illumina HiSeq 2000. Sequence reads were mapped to the hg19 reference genome using BWA[16]. Variant calling and quality control were performed as previously described[21]. A count for the total number of reads in a sample was scraped from the NCBI sequence read archive using the R package RCurl[22] while reads aligned to the mitochondrial genome were downloaded directly through Samtools (version 1.3.1). A raw measure of mtDNA-CN was calculated as the ratio of mitochondrial reads to the number of total aligned reads. Unlike WES, we did not observe large batch effects in our WGS raw mtDNA-CN calls, obviating the need for adjustment for Picard sequencing summary metrics.

**Digital PCR.**   mtDNA-CN was calculated using a multiplexed digital plate-based PCR (dPCR) method utilizing the *ND1* and *RPPH1* qPCR probes previously described. Samples were divided into 36,000 partitions on a 24-well plate and the fluorescence for each probe was measured with the Constellation Digital PCR System (Formulatrix, Boston MA). Fluorescence intensity was evaluated with the Formulatrix software and thresholds were based on visual inspection of the aggregate data for each plate. Thresholds were then used to determine the number of positive and negative partitions. Positive counts were fitted to a Poisson distribution to determine copy number[23]. mtDNA-CN was represented as the ratio between the number of *ND1* copies/μL and the number of *RPPH1* copies/μl. Samples were included if they had fewer than 30,000 positives for *ND1* and between 5 and 2,000 positives for *RPPH1*. Samples were filtered if the observed ratio was not between 15 and 300 *ND1:RPPH1*. The initial mtDNA-CN ratio was adjusted for plate as a random effect to represent a raw absolute measure of mtDNA-CN.

## Cardiovascular disease definition and adjudication

Event adjudication through 2017 in ARIC and 2015 in MESA consisted of expert committee review of death certificates, hospital records and telephone interviews. Incident cardiovascular disease (CVD) was defined as either incident coronary artery disease (CAD) or incident stroke. Incident CAD was defined as first incident MI or death owing to CAD while incident stroke was defined as first nonfatal stroke or death due to stroke. Individuals in ARIC with prevalent CVD at baseline were excluded from incident analyses.

## Genotyping and imputation

Genotype calling for the WBC count locus was derived from the Affymetrix Genome-wide Human SNP Array 6.0 in ARIC and MESA. Haplotype phasing for both cohorts was

performed using ShapeIt[24] and imputation was performed using IMPUTE2[25]. Genotypes were imputed to the 1000G reference panel (Phase I, version 3). Imputation quality for the Duffy locus lead SNP (rs2814778) was 0.95 and 0.92 in ARIC and MESA, respectively.

### DNA extraction method

All DNA used in the DNA extraction comparison were derived from HEK293T cells grown in a single 150T flask to minimize variation due to clonality and cell culture procedures. Extraction were performed with 15 replicates each containing one million cells. mtDNA-CN was determined using qPCR as described previously. To account for the inherent variability in mtDNA-CN estimation, qPCR was run in triplicate.

### Silica-based column extraction

We performed a silica-based column extraction using the AllPrep DNA/RNA Mini Kit (Qiagen) according to the manufacturer's instructions for fewer than $5 \times 10^6$ cells. Briefly, HEK293T cells were lysed and the subsequent lysate was pipetted directly onto the DNA Allprep spin column for homogenization and DNA binding. The bound DNA was then washed and eluted.

### Organic solvent extraction

An aliquot of cells were lysed with 350 μL of RLT Plus Buffer (Qiagen) and one volume of phenol:chloroform:isoamyl alcohol (25:24:1) (PCIAA) was added to the sample and mixed until it turned milky white. The solution was centrifuged and the upper aqueous phase containing DNA was transferred to a separate tube. We proceeded with an ethanol precipitation protocol using 3M sodium acetate to complete the DNA extraction.

### Direct cell lysis

Cells were pelleted at 500 g for 5 minutes and the supernatant was removed. The cell pellet was agitated in 100 μL of QuickExtract DNA Solution (Lucigen) to disrupt the pellet and placed in a thermocycler for 15 minutes at 68˚C followed by 10 minutes at 95˚C. The cell lysate was then centrifuged at 17,000 g for 15 minutes to pellet any insoluble inhibitors and the supernatant was transferred to a clean tube. The supernatant containing DNA was finally diluted 1:30 with water to limit the impact of any soluble inhibitors on qPCR.

### Statistical analyses

The final mtDNA-CN phenotype for all measurement techniques is represented as the standardized residuals from a linear model adjusting the raw measure of mtDNA-CN for age, sex, DNA collection center, and technical covariates. Additionally, mtDNA-CN in ARIC was adjusted for WBC count, and the14.9% of individuals with missing WBC data were imputed to the mean. WBC was not available in MESA for the same visit in which the DNA was obtained. As mtDNA-CN was standardized, the effect size estimates are in units of standard deviations, with positive betas corresponding to an increase in mtDNA-CN.

For analyses involving outcomes which also served as covariates in our final phenotype model (age, sex, WBC count), mtDNA-CN was calculated using the full model minus the outcome variable. For example, when exploring the relationship between mtDNA-CN and age, our mtDNA-CN phenotype would represent the standardized residuals from a model controlling for sex, sample collection center, WBC count and any technical covariates. We would then use this phenotype to explore the association between age and mtDNA-CN such that effect sizes for all comparisons remain in standard deviation units.

The Duffy locus is highly associated with WBC count in Blacks[26] due to its role in conferring a selective advantage to malaria, however this association is limited or absent in other ethnicities[27]. As such, single SNP regression for mtDNA-CN on the Duffy locus was limited to Blacks. Due to the association of mtDNA-CN with WBC count, the Duffy locus acts as another independent external validator for mtDNA-CN unadjusted for WBC count. In ARIC, mtDNA-CN not adjusted for WBC count was used as the independent variable. Single SNP regression models were additionally adjusted for age, sex, sample collection site, and genotyping PCs. Regression analyses were performed with FAST[28].

Cox-proportional hazards regression was used to estimate hazard ratios (HRs) for incident CVD outcomes. Follow-up time was defined from DNA collection through death, lost to follow-up, or study end point (through 2017 in ARIC and 2015 in MESA).

Pairwise F-tests were used to test the null hypothesis that the ratio of variances between the DNA extraction methods is equal to one.

All statistical analyses were performed using R (version 3.3.3).

## Ethics statement

Johns Hopkins IRB approved of this study (NA_00091014 / CR00027367). All participants provided written informed consent and all centers obtained approval from their respective institutional review boards.

## Results

The study included 1,085 participants from ARIC with mtDNA-CN data from the Affymetrix 6.0 microarray, the Illumina Exome Chip microarray, WES, and WGS while MESA included 3,489 participants with mtDNA-CN data available from qPCR, the Affymetrix 6.0 microarray, and the Illumina Exome Chip microarray (combined N = 4,574). The mean age of study participants was 61.4 years (ARIC, 57.1 years; MESA 62.7 years), 55.3% of participants were female (n = 2,528), and 46.4% of participants were Black (n = 2,124) (Table 1). While the Affymetrix and Illumina Exome Chip arrays were run in both cohorts, at the time of analysis WES and WGS were unique to ARIC and qPCR was unique to MESA.

### mtDNA-CN estimation method comparison

To determine the optimal method for measuring mtDNA-CN, we ranked the performance of each technique based on strength of the association, as measured by *p* values, with the relevant mtDNA-CN correlate (S2 Table). Kendall's W tests[29] show significant agreement in

**Table 1. Participant characteristics.**

| Participant Characteristics | ARIC | MESA |
|---|---|---|
| N | 1,085 | 3,489 |
| Sex (female) | 672 (61.9) | 1,856 (53.2) |
| Ethnicity (Black) | 958 (88.3) | 1,226 (35.1) |
| Age | 57.1 ± 5.9 | 62.7 ± 10.2 |
| WBC count ($10^3$/μl) | 5.8 ± 1.7 | NA |
| Incident CVD | 174 (16.0) | 270 (7.7) |

Values are number (%) or mean ± SD;

Abbreviations: SD, standard deviation; WBC, white blood cell;

CVD, cardiovascular disease

**Table 2. Performance rankings for mtDNA-CN estimation methods.**

| Cohort | Assay | Age | Sex | WBC | Duffy locus* | Incident CVD | Mean Rank | Kendall's W *p* value |
|--------|-------|-----|-----|-----|--------------|--------------|-----------|----------------------|
| ARIC | Exome | 2 | 4 | 3 | 4 | 4 | 3.4 | 0.001 |
| | Affy | 3 | 2 | 2 | 2 | 2 | 2.2 | |
| | WES | 4 | 3 | 4 | 3 | 3 | 3.4 | |
| | WGS | 1 | 1 | 1 | 1 | 1 | 1 | |
| MESA | Exome | 2 | 3 | NA | 2.5 | 3 | 2.625 | 0.03 |
| | Affy | 1 | 1 | NA | 1 | 1 | 1 | |
| | qPCR | 3 | 2 | NA | 2.5 | 2 | 2.375 | |

*Duffy locus associations were performed in Blacks only

rankings across correlates in ARIC (*p* = 0.0019, Kendall's W = 0.79) and MESA (*p* = 0.036, Kendall's W = 0.82) with WGS and the Affymetrix array performing best for each measure in ARIC and MESA, respectively (Table 2).

To additionally quantify performance, we created a scoring system for each method using negative log transformed *p* values standardized to the least significant method for each correlate. These values were then summed across the correlates for each method to achieve an overall rating of performance (S3 Table). These ratings were compared to 1,000 permutations of a random sampling of the standardized and transformed *p* values for each correlate across the different estimation techniques. In ARIC, WGS had a significantly higher performance score compared to all other methods (*p* < 0.002) while the Illumina Exome Chip had a significantly lower score (*p* = 0.03) (S1A Fig). In MESA, Affymetrix had a significantly higher score than qPCR and the Illumina Exome Chip (*p* = 0.002) (S1B Fig). When removing the contribution of WGS in ARIC, the Affymetrix array had a significantly higher score than the Illumina Exome Chip and WES (*p* = 0.01) (S1C Fig).

As WGS and Affymetrix performed similarly, we sought to further parse out their performance by evaluating the 2,746 ARIC samples which contained mtDNA-CN from both platforms. On average, WGS performed 2.2 orders of magnitude more significantly than the Affymetrix array (S4 Table).

Due to the recent emergence of digital PCR (dPCR) as a viable method for calculating mtDNA-CN, we performed an additional exploratory analysis in 983 individuals of the MESA cohort comparing the performance of dPCR to qPCR and the Affymetrix genotyping array (S5 Table). While mtDNA-CN calculated from dPCR was more significantly associated with age then either qPCR or the Affymetrix array, dPCR was the least significantly associated metric with sex and the observed association with incident CVD was in the opposite direction as expected (S6 Table).

## DNA extraction comparison

Raw mitochondrial estimates from qPCR were mean-zeroed to the plate average and the mean value across the triplicate plates was used to determine the variance across the 15 replicates for each method (Fig 1). The variance for our novel Lyse method was significantly lower at 0.02 compared to 0.17 and 0.59 for the PCIAA and Qiagen Kit extractions respectively (F = 0.13, *p* = 5.44x10$^{-4}$; F = 0.04, *p* = 2.82x10$^{-7}$). Additionally, our findings support previous work[14] demonstrating PCIAA had significantly lower variability compared to the Qiagen Kit (F = 0.29, *p* = 0.03).

## Discussion

We explored several methods for measuring mtDNA-CN in 4,574 self-identified White and Black participants from the ARIC and MESA studies. We found mtDNA-CN estimated from

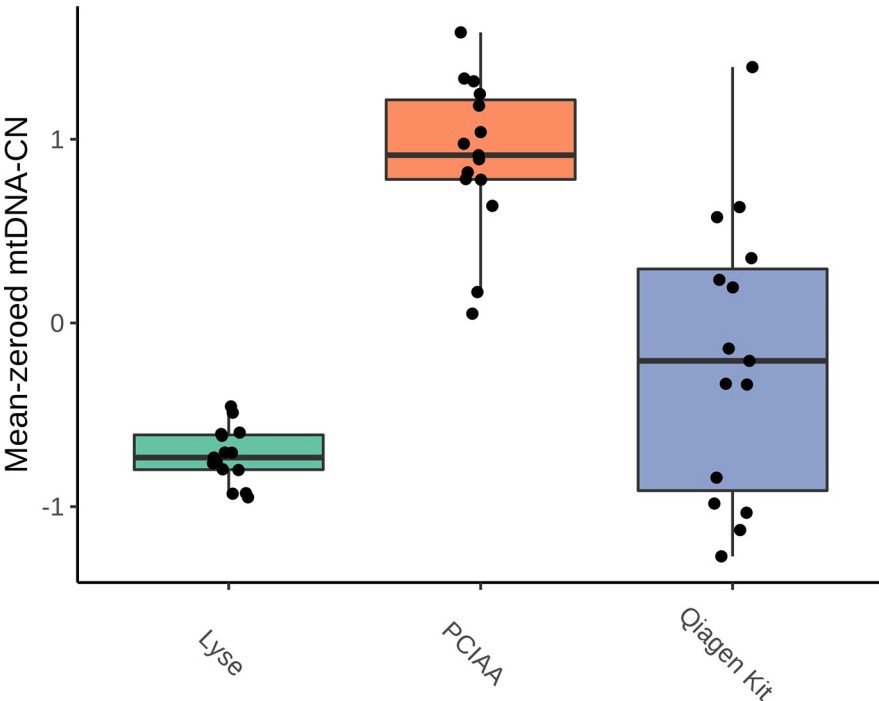

**Fig 1. mtDNA-CN measured across DNA extraction methods.** mtDNA-CN measured by qPCR was mean-zeroed and averaged across three runs for Lyse, PCIAA and Qiagen Kit DNA extractions. Variance for Lyse, PCIAA and Qiagen Kit are 0.02, 0.17 and 0.59 respectively. PCIAA, phenol:chloroform:isoamyl alcohol.

WGS read counts and Affymetrix Genome-Wide Human SNP Array 6.0 probe intensities was more significantly associated with known mtDNA-CN correlates compared to mtDNA-CN estimated from WES, qPCR and the Illumina HumanExome BeadChip. When observing the relative performance of these methods, mtDNA-CN calculated from either WGS or Affymetrix array are, respectively, 5.6 and 5.4 orders of magnitude more significant than the current gold standard of qPCR (Fig 2). These results are not limited to significance as we see similar trends when exploring effect size estimates (Fig 3). For example, when looking at incident CVD, mtDNA-CN measured from WGS observes a substantial HR of 0.63 (0.54–0.74) where as

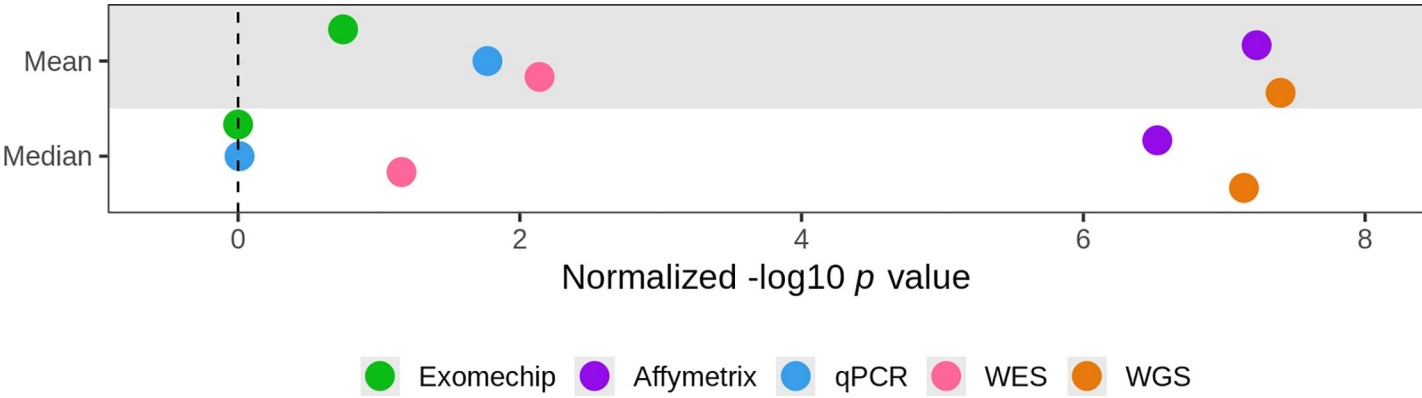

**Fig 2. Relative overall performance of mtDNA-CN estimation methods.** Overall performance for each method scored as mean or median of the negative log-transformed *p* value across all correlates normalized to the least significant method of each correlate. For ExomeChip and Affymetrix, the mean value across both cohorts was used as the final measure of performance.

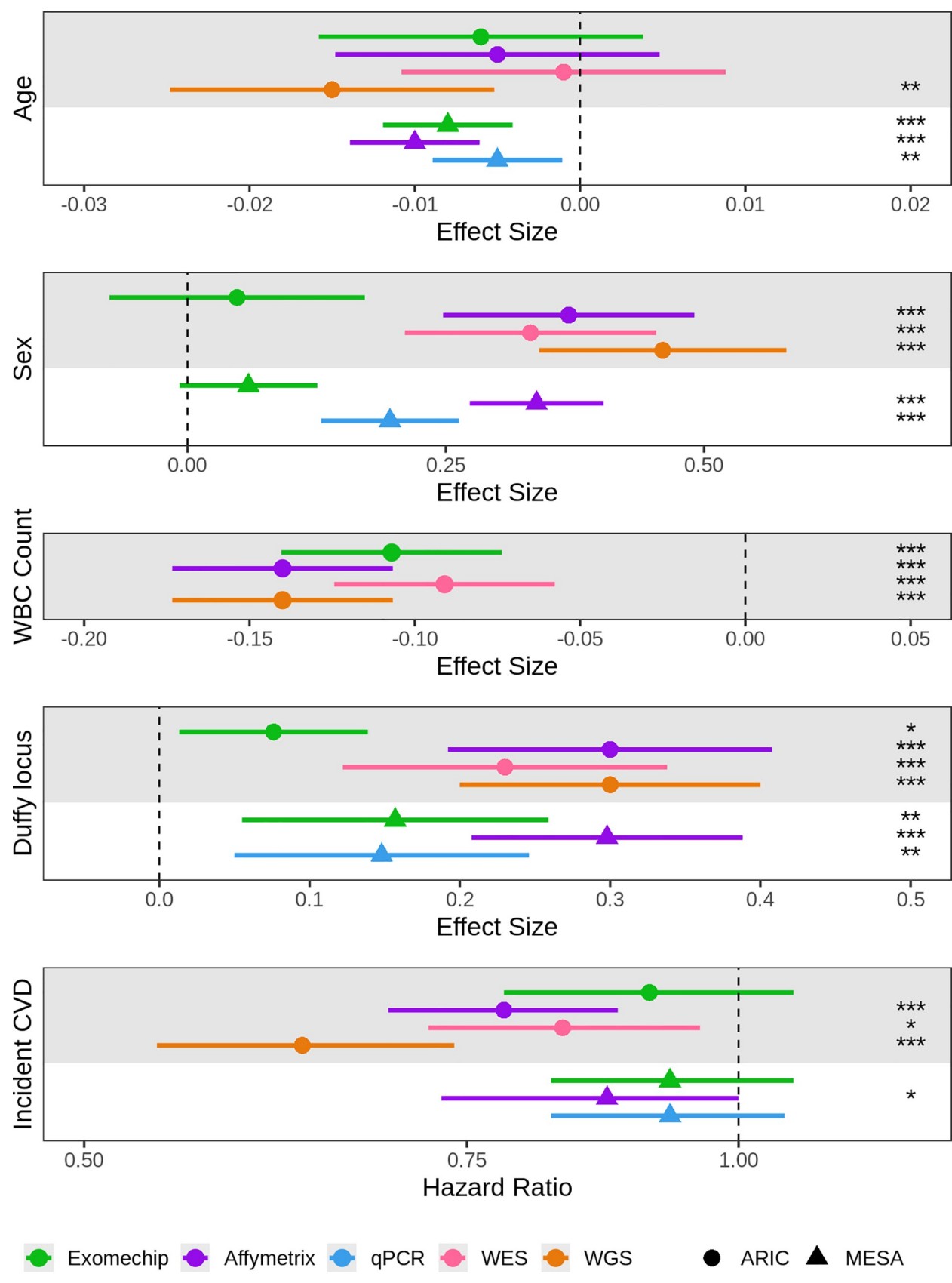

**Fig 3. Effect size and hazard ratio estimates for mtDNA-CN with known correlates.** Data points and their corresponding 95% confidence intervals represent the effect size or hazard ratio estimates for mtDNA-CN with Age, Sex, white blood cell (WBC) count, Duffy locus, and incident cardiovascular disease (CVD). Effect size estimates are in standard deviation units. The significance of each estimate is represented as '*' for $P < 0.05$, '**' for $P < 0.01$, and '***' for $P < 0.001$. WBC, white blood cell.

mtDNA-CN measured from qPCR only has a HR of 0.93 (0.82–1.05), a marked difference. As a result, when exploring the relationship between mtDNA-CN and a trait of interest, on average one could expect a result 5.6 orders of magnitude less significant and 6 times less extreme when using mtDNA-CN estimated from qPCR data as opposed to WGS.

Several recent reports have touted dPCR as the new gold standard for mtDNA-CN estimation due to its ability to quantify absolute copy number[30–32]. In a small subset of MESA samples, we found mtDNA-CN estimates from dPCR were on average 1.15 and 0.55 orders of magnitude less significant than Affymetrix and qPCR respectively (S7 Table). These results suggest dPCR may not measure mtDNA-CN as accurately as both the current gold standard and other recently developed methods. However, it is important to note these findings were derived from a subset of samples a fifth of the size as those from the main findings of the overall study, and thus should be interpreted with caution. Additionally, whereas the dPCR data was derived from DNA from packed red blood cells, the qPCR and Affymetrix data was obtained from peripheral leukocytes potentially explaining the poor performance of dPCR relative to other methods.

Interestingly, mtDNA-CN measured from two seemingly similar microarray platforms differed drastically (S2 Fig). However, this finding is unsurprising when exploring the underlying biochemistry of sample preparation for each microarray platform. While the Affymetrix protocol starts with two restriction enzyme digests prior to whole genome amplification (WGA), the Illumina Exome Chip requires WGA with a processive polymerase prior to sonication. As a result, the mitochondrial genome undergoes rolling circle amplification which occurs at a significantly faster rate than linear WGA[33].

Lower mtDNA-CN has been found to be associated with an increased incidence for several diseases, including end stage renal disease, type 2 diabetes, and non-alcoholic fatty liver disease[34–36]. However, such studies have relied on mtDNA-CN estimated from qPCR data. Our findings suggest much of the current literature may be severely underestimating disease associations with mtDNA-CN as well as its potential as a predictor of disease outcomes. Despite this, at <\$2 per sample qPCR may remain the principal method for measuring mtDNA-CN due to the prohibitive costs of WGS. Furthermore, absolute quantification of mtDNA-CN through the use of standard curves may improve upon the performance of qPCR furthering its continuing use[37].

We additionally showed DNA extraction method affects mtDNA-CN estimate reproducibility with copy number measured directly from cell lysate significantly outperforming silica-based column extraction and organic solvent extraction. Although several other studies have explored the impact of DNA isolation protocol on mtDNA-CN estimation[14,38,39], to our knowledge, this is the first study to interrogate the possibility of measuring mtDNA-CN directly from cell lysate. In addition to the superior performance of direct cell lysis, this method is cheaper and has less hands-on time than PCIAA or Qiagen Kit extractions. However, the authors recognize DNA from cell lysate has less downstream utility than traditional DNA extraction procedures potentially limiting its adoption within the mtDNA-CN field when sample availability is limited. Additionally, as our application of the lyse method was limited to cultured cells, it is important to further validate this method in the context of different sample types which may have higher concentrations of inhibitors. Furthermore, it is important to note the various DNA extraction methods resulted in significantly different

mtDNA-CN estimates ($p = 3.56 \times 10^{-11}$, 0.02, $2.85 \times 10^{-7}$ for Lyse:PCIAA, Lyse:Qiagen Kit, and PCIAA:Qiagen Kit respectively). As such, when choosing an extraction method, it is important to remain consistent across the study.

In conclusion, our study demonstrates mtDNA-CN calculated from WGS reads or Affymetrix microarray probe intensities significantly improves upon the current gold standard method of qPCR. Furthermore, we show direct cell lysis introduces less variability to mtDNA-CN estimates than popular DNA extraction methods. Despite the relative infancy of using mtDNA-CN as a novel risk marker, these findings highlight the need for the field to adapt to current technologies to ensure disease and trait associations are fully realized with a move toward more accurate microarray and WGS methods. Furthermore, due to the prevalence of qPCR in the literature, the authors recommend re-analyzing trait associations as more WGS data becomes available from large initiatives such as TOPMed.

## Supporting information

**S1 Fig. Permutation test for mtDNA-CN estimation method performance.**
(TIF)

**S2 Fig. Phenotype correlation plots.**
(TIF)

**S1 Table. Picard sequencing summary metrics definitions.**
(XLSX)

**S2 Table. Associations of known correlates with mtDNA-CN estimation platforms.** *Duffy locus associations were performed in Blacks only.
(XLSX)

**S3 Table. Relative performance of methods as rated by standardized -log *p* values.** *Duffy locus associations were performed in Blacks only.
(XLSX)

**S4 Table. Relative performance of WGS and Affymetrix as rated by standardized -log *p* values.** *Duffy locus associations were performed in Blacks only.
(XLSX)

**S5 Table. Participant characteristics for dPCR subset.** Values are number (%) or mean ± SD; Abbreviations: SD, standard deviation; CVD, cardiovascular disease.
(XLSX)

**S6 Table. Associations of known correlates with mtDNA-CN estimation platforms for dPCR subset.** *Duffy locus associations were performed in blacks only.
(XLSX)

**S7 Table. Relative performance of methods as rated by standardized -log *p* values for dPCR subset.** *Duffy locus associations were performed in blacks only. +Affymetrix and dPCR effect size estimates were in opposite direction as known effects and thus the -log p value of qPCR was standardized to a p value of 1 for Affymetrix and dPCR.
(XLSX)

## Acknowledgments

We thank the staff and participants of the Atherosclerosis Risk in Communities Study, Cardiovascular Health Study, and the Multi-Ethnic Study of Atherosclerosis studies for their

important contributions. A full list of participating MESA investigators and institutions can be found at http://www.mesa-nhlbi.org. Digital PCR was conducted at the Genetic Resources Core Facility, Johns Hopkins Institute of Genetic Medicine, Baltimore, MD.

## Author Contributions

**Conceptualization:** Ryan J. Longchamps, Eliseo Guallar, Nathan Pankratz, Dan E. Arking.

**Data curation:** Ryan J. Longchamps, Christina A. Castellani, Stephanie Y. Yang, Charles E. Newcomb, Jason A. Sumpter, Megan L. Grove, Kent D. Taylor, Jerome I. Rotter, Eric Boerwinkle, Dan E. Arking.

**Formal analysis:** Ryan J. Longchamps, Christina A. Castellani, Stephanie Y. Yang, Charles E. Newcomb, Jason A. Sumpter, John Lane, Dan E. Arking.

**Funding acquisition:** Eliseo Guallar, Kent D. Taylor, Jerome I. Rotter, Eric Boerwinkle, Dan E. Arking.

**Project administration:** Megan L. Grove.

**Resources:** Megan L. Grove.

**Software:** John Lane.

**Supervision:** Eliseo Guallar, Nathan Pankratz, Jerome I. Rotter, Eric Boerwinkle, Dan E. Arking.

**Writing – original draft:** Ryan J. Longchamps, Dan E. Arking.

**Writing – review & editing:** Ryan J. Longchamps, Christina A. Castellani, Stephanie Y. Yang, Charles E. Newcomb, Jason A. Sumpter, John Lane, Megan L. Grove, Eliseo Guallar, Nathan Pankratz, Kent D. Taylor, Jerome I. Rotter, Eric Boerwinkle, Dan E. Arking.

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
