## [Decision Letter · Decision Letter 0]

26 Nov 2019

PONE-D-19-28415

Evaluation of mitochondrial DNA copy number estimation

PLOS ONE

Dear Dr. Arking,

Thank you for submitting your manuscript to PLOS ONE. After careful consideration, we feel that it has merit but does not fully meet PLOS ONE’s publication criteria as it currently stands. Therefore, we invite you to submit a revised version of the manuscript that addresses the points raised during the review process.

We would appreciate receiving your revised manuscript by Jan 10 2020 11:59PM. To enhance the reproducibility of your results, we recommend that if applicable you deposit your laboratory protocols in protocols.io, where a protocol can be assigned its own identifier (DOI) such that it can be cited independently in the future. For instructions see: http://journals.plos.org/plosone/s/submission-guidelines#loc-laboratory-protocols

We look forward to receiving your revised manuscript.

Kind regards,

David C. Samuels

Academic Editor

PLOS ONE

Journal Requirements:

a) Please provide an amended Funding Statement that declares *all* the funding or sources of support received during this specific study (whether external or internal to your organization) as detailed online in our guide for authors at http://journals.plos.org/plosone/s/submit-now.  

b) Please state what role the funders took in the study.  If any authors received a salary from any of your funders, please state which authors and which funder. If the funders had no role, please state: "The funders had no role in study design, data collection and analysis, decision to publish, or preparation of the manuscript."

5. Your ethics statement must appear in the Methods section of your manuscript. If your ethics statement is written in any section besides the Methods, please move it to the Methods section and delete it from any other section. Please also ensure that your ethics statement is included in your manuscript, as the ethics section of your online submission will not be published alongside your manuscript.

Reviewers' comments:

Reviewer's Responses to Questions

**Comments to the Author**

1. Is the manuscript technically sound, and do the data support the conclusions?

Reviewer #1: Yes

2. Has the statistical analysis been performed appropriately and rigorously? 

Reviewer #1: Yes

3. Have the authors made all data underlying the findings in their manuscript fully available?

Reviewer #1: No

4. Is the manuscript presented in an intelligible fashion and written in standard English?

Reviewer #1: Yes

5. Review Comments to the Author

Reviewer #1: Lonchamps et al. provide their thorough work in evaluating several methods for assessing mtDNA-CN. Overall, the paper is well-written and provides robust data analysis methods. I recommend minor revisions to address the following points.

• Can the authors please either (1) provide and explanation for why the Duffy locus was only analyzed in African Americans, or (2) provide citation(s) as justification?

• Figure 3- the legend is seemingly mislabeled; I believe that the green series represents the ExomeChip data and the Purple series represents Affy.

• Given that the authors only tested the “Lyse” method on cultured cells, which likely have fewer inhibitors compared to other sample types (e.g., heme carry over in blood derived samples), the authors should discuss briefly that the method may have limited applicability based on sample type.

• The authors discuss and evaluate relative qPCR for quantification of mtDNA-CN, but do not evaluate or discuss any of the published absolute qPCR assays; this may warrant a brief discussion point since absolute qPCR has the advantage of monitoring PCR efficiency (which can greatly alter copy number estimates) and batch effects.

• Throughout the manuscript, the authors interchangeably use "black" and "African Americans"; I would suggest picking one or the other to maintain consistency.

• If the authors choose to use "white" and "black" as their racial descriptors, please capitalize the first letter of each term.

• Pg.10: "DNA for mtDNA-CN estimation was collected from different visits and was derived from buffy coat using the Gentra Puregene Blood Kit (Qiagen)."---Can the authors verify that relevant variables (e.g., WBC count) were also collected from the same visit as the DNA sample?

• Pg.16: "Follow-up time was defined from DNA collection through death, loss to follow-up, or study end point (through 2017 in ARIC and 2015 in MESA)."—suggested edit to “lost to follow-up”.

• Many of the tables have poor resolution or illegible text; particularly, in supplemental Figure 2, the authors should consider changing the x and y axis labels to a larger font size.

6. PLOS authors have the option to publish the peer review history of their article (what does this mean?). If published, this will include your full peer review and any attached files.

Reviewer #1: No

---

## [Author Response · Author response to Decision Letter 0]

30 Dec 2019

We wish to thank the reviewer for their thorough and helpful review of our manuscript entitled “Evaluation of mitochondrial DNA copy number estimation techniques”. We have addressed the reviewer comments in the revised manuscript and listed our responses to the comments below. Reviewer comments are listed in red text, our response is shown in black and text which was added to our manuscript is italicized in quote blocks.

 Comments:

1. Can the authors please either (1) provide and explanation for why the Duffy locus was only analyzed in African Americans, or (2) provide citation(s) as justification?

We agree with the reviewer that we were not clear as to why the Duffy locus was 1) chosen as a correlate and 2) why it was only analyzed in Blacks. We have provided the following additional information between lines 259 and 263 on page 13 within the statistical analyses section of the Methods. 

“The Duffy locus is highly associated with WBC count in Blacks26 due to its role in conferring a selective advantage to malaria, however this association is limited or absent in other ethnicities27. As such, single SNP regression for mtDNA-CN on the Duffy locus was limited to Blacks. Due to the association of mtDNA-CN with WBC count, the Duffy locus acts as another independent external validator for mtDNA-CN unadjusted for WBC count.”

2. Figure 3- the legend is seemingly mislabeled; I believe that the green series represents the ExomeChip data and the Purple series represents Affy.

Thank you, the figure has been modified.

3. Given that the authors only tested the “Lyse” method on cultured cells, which likely have fewer inhibitors compared to other sample types (e.g., heme carry over in blood derived samples), the authors should discuss briefly that the method may have limited applicability based on sample type.

We have added the following comment on page 19 (lines 416-419) within our discussion to highlight that our Lyse method needs further validation in non-cultured cells. 

“Additionally, as our application of the lyse method was limited to cultured cells, it is important to further validate this method in the context of different sample types which may have higher concentrations of inhibitors.”

4. The authors discuss and evaluate relative qPCR for quantification of mtDNA-CN, but do not evaluate or discuss any of the published absolute qPCR assays; this may warrant a brief discussion point since absolute qPCR has the advantage of monitoring PCR efficiency (which can greatly alter copy number estimates) and batch effects.

We agree and have added a comment on page 19 of the discussion (lines 404-407) pointing out that absolute qPCR may improve upon the performance of mtDNA-CN estimation.

“Furthermore, absolute quantification of mtDNA-CN through the use of standard curves may improve upon the performance of qPCR furthering its continuing use37.”

5. Throughout the manuscript, the authors interchangeably use "black" and "African Americans"; I would suggest picking one or the other to maintain consistency.

Thank you, this change has been made as suggested

6. If the authors choose to use "white" and "black" as their racial descriptors, please capitalize the first letter of each term.

Thank you, we have capitalized all racial descriptors.

7. Pg.10: "DNA for mtDNA-CN estimation was collected from different visits and was derived from buffy coat using the Gentra Puregene Blood Kit (Qiagen)."---Can the authors verify that relevant variables (e.g., WBC count) were also collected from the same visit as the DNA sample?

We have clarified that relevant covariates were derived from the same visit in which the DNA was collected.

8. Pg.16: "Follow-up time was defined from DNA collection through death, loss to follow-up, or study end point (through 2017 in ARIC and 2015 in MESA)."—suggested edit to “lost to follow-up”.

The suggested edits have been made.

9. Many of the tables have poor resolution or illegible text; particularly, in supplemental Figure 2, the authors should consider changing the x and y axis labels to a larger font size.

Thank you for pointing this out, the figures and tables have been adjusted.

During the review process we were additionally made aware that the DNA used in our dPCR exploratory study was derived from packed red blood cells and not peripheral leukocytes as we previously believed. We have made this clear on page 6 of the methods as well as commented briefly on this difference on page 18 of the discussion the discussion.

“Additionally, whereas the dPCR data was derived from DNA from packed red blood cells, the qPCR and Affymetrix data was obtained from peripheral leukocytes potentially explaining the poor performance of dPCR relative to other methods.”

We thank you for your time and consideration of our revised manuscript. We look forward to your feedback.

Yours sincerely,

Dan E. Arking, PhD

---

## [Editor Report · Decision Letter 1]

9 Jan 2020

Evaluation of mitochondrial DNA copy number estimation techniques

PONE-D-19-28415R1

Dear Dr. Arking,

We are pleased to inform you that your manuscript has been judged scientifically suitable for publication and will be formally accepted for publication once it complies with all outstanding technical requirements.

With kind regards,

David C. Samuels

Academic Editor

PLOS ONE
---

## [Editor Report · Acceptance letter]

14 Jan 2020

PONE-D-19-28415R1 

Evaluation of mitochondrial DNA copy number estimation techniques 

Dear Dr. Arking:

I am pleased to inform you that your manuscript has been deemed suitable for publication in PLOS ONE. Congratulations! Your manuscript is now with our production department. 

With kind regards,

on behalf of

Dr. David C. Samuels 

Academic Editor

PLOS ONE